

# Mixmasters in Wonderland: Chaotic dynamics from Carroll limits of gravity

Gerben Oling[1] and Juan F. Pedraza[2]

**1** School of Mathematics and Maxwell Institute for Mathematical Sciences,
University of Edinburgh, Peter Guthrie Tait Road, Edinburgh EH9 3FD, UK
**2** Instituto de Física Teórica UAM/CSIC, Calle Nicolás Cabrera 13-15, Madrid 28049, Spain

## Abstract

We demonstrate that the Carroll limit of general relativity coupled to matter captures the chaotic mixmaster dynamics of near-singularity limits. Zooming in on the behavior of general relativity close to spacelike singularities reveals rich and solvable ultra-local Belinski-Khalatnikov-Lifshitz (BKL) dynamics, which we show to be captured by a Carroll limit. Specifically, building on recent work on geometric Carroll expansions of general relativity, we establish that leading-order Carroll gravity, with suitable matter coupling, accurately describes well-known cosmological billiards behavior. Since the Carroll limit implements the ultra-local limit off shell, this opens up the door to a wide range of possible tractable applications, including spatially inhomogeneous setups and the emergence of spikes at late times. This further suggests that Carroll gravity, along with its subleading corrections, could serve as a valuable tool for studying deep infrared physics in AdS/CFT.

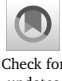

# 1 Introduction

It has long been known that general relativity can lead to rich yet tractable behavior in the vicinity of spacelike singularities [1,2]. Roughly speaking, spatial derivatives are suppressed in such Belinski-Khalatnikov-Lifshitz (BKL) near-singularity limits, leading to emerging ultra-local behavior. The archetypical solution of Einstein's equations in this limit is given by the Kasner metric,

$$ds^2 = -dt^2 + t^{2p_1}dx^2 + t^{2p_2}dy^2 + t^{2p_3}dz^2 \,, \tag{1}$$

which describes a spatially homogeneous geometry that expands anisotropically with fixed scaling exponents. This metric is Einstein if the scaling exponents $p_a$ satisfy particular relations. In the absence of matter, these relations are

$$\sum_a p_a = 1 \,, \qquad \sum_a (p_a)^2 = 1 \,. \tag{2}$$

The near-singularity region of highly symmetric black hole solutions are described by a single Kasner geometry, corresponding to a fixed set of values of the scaling exponents. The rich dynamics of BKL limits arises in situations where the Kasner exponents vary dynamically through the space of solutions that is parametrized by relations such as (2).

This dynamics can be sourced in several different ways. First, we can modify the Kasner Ansatz (1) by adding curvature to its spatial slices. To obtain a curved metric whose spatial slices are homogeneous but anisotropic, it is useful to start from a thee-dimensional group manifold such as $SO(3)$, whose natural metric is both homogeneous and isotropic. Anisotropy can then be introduced using scaling exponents similar to the $p_a$ in the Kasner metric (1), and this is known as the 'mixmaster' model [3]. As we briefly review in Section 2.2 below, the presence of spatial curvature introduces a potential in the space of scaling exponents, leading to rich and chaotic dynamics.

Additionally, interesting dynamics can be obtained by adding matter couplings. Notably, the idea of BKL limits was revisited some time ago in the context of of supergravity, where the $p$-form couplings were shown to lead to chaotic dynamics characterized by affine Lie algebras, as reviewed in [2,4]. These rich symmetry structures arise naturally when the dynamics in the space of scaling exponents is mapped to a particle moving in an external hyperbolic geometry, an idea going back to Chitre and Misner [5,6]. This particle motion is constrained by potentials which are determined by the setup at hand. At late times, the potentials lead to sharp walls, resulting in a 'cosmological billiard motion' with reflections at the walls. For example, the four-dimensional mixmaster model mentioned above maps to billiard dynamics in a two-dimensional hyperbolic triangle, corresponding to the $SL(2,\mathbb{Z})$ fundamental domain. In more intricate supergravity settings, the billiard table can be seen as the Weyl chamber of an affine Lie algebra [4].

More recently, the BKL phenomenon has been revisited through the lens of the AdS/CFT correspondence. To see how this may arise, let us first work out explicitly how a planar AdS-Schwarzschild black hole in 3+1 dimensions gives rise to a Kasner geometry near its singularity. Far behind the horizon, $z \gg z_H$, the metric becomes

$$ds^2 = \frac{L^2}{z^2}\left[ -\left(1-(z/z_H)^3\right)dt^2 + \frac{dz^2}{1-(z/z_H)^3} + dx^2 + dy^2 \right] \tag{3}$$

$$\approx -d\tau^2 + \tau^{-2/3}dt^2 + \tau^{4/3}\left[dx^2 + dy^2\right], \tag{4}$$

where we have introduced a new 'interior time' coordinate $\tau = \tau(z)$ and we have rescaled the remaining coordinates. This corresponds to a Kasner metric of the form (1) with $p_t = -1/3$ and $p_x = p_y = 2/3$, which satisfy the conditions (2) above.

While the exterior geometry of these AdS-Schwarzschild black holes is dynamically stable, their interiors are notoriously unstable. Matter fields experience infinite growth as they approach a spacelike singularity, causing significant backreaction [7, 8]. Generally speaking, the Schwarzschild singularity is said to be finely tuned within the range of potential near-singularity behaviors, making it an unusual late-time solution. Therefore, this inherent instability of the Schwarzschild singularity must be carefully considered in any holographic investigation of the black hole interior.

Motivated by this conceptual challenge, Frenkel, Hartnoll, Kruthoff and Shi [9] investigated a class of AdS black holes obtained by deforming the dual CFT with a relevant scalar operator, finding that such a deformation leads to the emergence of Kasner geometries other than (4) as the endpoint of the interior's evolution. Subsequent studies with different types of deformations [10–29] have revealed a plethora of rich near-singularity dynamics similar to the cosmological billiards discussed previously, depending on the matter content and interactions within the gravitational theory. In particular, this includes BKL-like phenomena such as Kasner inversions, the rapid collapse or expansion of the Einstein-Rosen bridge, and finite or infinite series of bounces similar to those seen in cosmological billiards. Holographically, there have been attempts to interpret these phenomena in terms of RG flows of the boundary CFT [30], and several field theory observables have been proposed to capture specific imprints of the BKL-type dynamics of the black hole interior [27–38].

Despite this recent progress, there is still no firm understanding of how the near-singularity Kasner exponents and their potentially chaotic dynamics arise from a boundary perspective. In order to isolate the relevant bulk dynamics, and to be able to fully explore its possibilities, building on earlier observations in [4, 39], we develop a novel approach to the near-singularity dynamics of general relativity coupled to matter, focusing on the ultra-local structure that arises naturally at late interior times. In this limit, the light cone collapses to a line, which corresponds to a 'small speed of light' contraction of the Lorentz algebra that leads to the Carroll algebra [40, 41]. Geometrically, the Carroll limit can be described in terms of fully covariant curved spacetime geometry using a toolkit similar to the Newton–Cartan geometry associated to non-relativistic limits, as recently reviewed in [42].

Over the past years, there has been a surge of interest in Carroll limits. This has mainly been motivated by flat space holography, where Carroll field theories naturally arise since the asymptotic BMS symmetries at null infinity can be interpreted as the symmetries of a conformal Carroll structure [43]. For this reason, conformal Carroll symmetries have been proposed as a guiding principle for holography in asymptotically flat spacetimes [44–46], complementing the celestial holography approach [47–49].

In this paper, however, we will use the ultra-local Carroll limit to describe *bulk* physics, focusing specifically on the near-singularity BKL dynamics of black holes. Our main aim will be to demonstrate explicitly that a theory of Carroll gravity, consisting of dynamical Carroll geometry coupled to matter, can describe the aforementioned mixmaster behavior. Moreover, this Carroll theory of gravity is directly obtained from an ultra-local limit of the bottom-up AdS/CFT model that was recently introduced in [23] to obtain mixmaster dynamics near the singularity of an asymptotically AdS black hole.

We want to argue that this approach to near-singularity dynamics is promising for several reasons. First, since the ultra-local limit is implemented off shell on the level of the geometry, the Carroll theory of gravity that we consider [50–53] is inherently much simpler than full general relativity (GR). In particular, as emphasized in [39, 50, 53, 54], its evolution equations always consist of *ordinary* differential equations, instead of the hyperbolic partial differential equations of GR. As such, our approach will allow us to construct a much larger class of models where the emergence of BKL-type behavior can be observed explicitly, either analytically or numerically, than what would be tractable in full GR. Additionally, the Carroll theory of gravity

that we use can be seen as the leading order in the ultra-local expansion of GR that was recently constructed in [53], building on a related construction of the off-shell non-relativistic expansion in [55–58]. While we only focus on the leading-order action in the present paper, our approach is therefore naturally equipped to explicitly studying subleading corrections to near-singularity BKL dynamics in a tractable way.

This paper is organized as follows. We start out by reviewing salient features of BKL dynamics in Einstein gravity. For this, we first introduce a useful parametrization of Kasner-type geometries with time-dependent scaling exponents in Section 2.1, and we show how it gives rise to the interpretation of such geometries as trajectories in a minisuperspace parametrized by the scaling exponents. In Section 2.2, we then introduce the mixmaster model, and we show how it leads to nontrivial trajectories, which can furthermore be mapped to billiard dynamics on a hyperbolic triangle. Section 3 then introduces the necessary Carroll tools, including a brief overview of curved Carroll geometry, the off-shell ultra-local expansion of general relativity introduced in [53] and the coupling to Carroll limits of matter actions. We then show in Section 4.1 that leading-order vacuum Carroll gravity admits Kasner solutions[1] and similarly leads to trajectories in an external Minkowski space. Motivated by the recently-proposed AdS/CFT model from [23], we then demonstrate explicitly in Section 4.2 that the Carroll limit of gravity coupled to three abelian gauge fields reproduces the full dynamics of the mixmaster model. Finally, we summarize our findings and set out several future directions in Section 5.

## 2 Kasner geometries and BKL in Einstein gravity

We first review some aspects of Kasner metrics and BKL dynamics in Einstein gravity. The rich dynamics that we are interested in arises when the Kasner exponents in (1) are allowed to vary in time. Additionally, it is convenient to introduce a lapse function. We therefore now consider metrics of the form

$$ds^2 = -e^{-2\alpha(t)}dt^2 + e^{2\beta_1(t)}dx^2 + e^{2\beta_2(t)}dy^2 + e^{2\beta_3(t)}dz^2. \tag{5}$$

From this, the vacuum Einstein equations reproduce Kasner geometries with fixed scaling exponents, as we will show in Section 2.1. Following for example [2], this parametrization naturally suggests an interpretation of such geometries as null geodesics in an external three-dimensional Minkowski 'minisuperspace' parametrized by the scaling exponents.

As depicted in Figure 1, a particular Kasner geometry then corresponds to a null line inside the light cone of this external Minkowski space, which maps to a point on a given hyperbolic slice inside this light cone. To illustrate how the rich and chaotic BKL-type dynamics arises, we briefly review the mixmaster model in Section 2.2. Here, we will see that adding spatial curvature introduces a potential in the space of allowed scaling exponents, leading to sharp walls at late times. This leads to rich and chaotic dynamics, which can be mapped to billiard dynamics of a particle on a hyperbolic slice (or any other spacelike slicing of the light cone interior).

### 2.1 Kasner geometries in vacuum

With the metric Ansatz (5), the $tt$ component of the vacuum Einstein equations implies

$$0 = -2\left(\dot{\beta}_1\dot{\beta}_2 + \dot{\beta}_1\dot{\beta}_3 + \dot{\beta}_2\dot{\beta}_3\right). \tag{6}$$

---

[1]The fact that Carroll gravity has Kasner-type solutions with fixed scaling exponents as in (1) was observed in [59,60] and was also pointed out by M. Henneaux during the 2022 Vienna Carroll Workshop. However, these references did not reproduce BKL or mixmaster dynamics from Carroll gravity. Our aim in the present paper is to demonstrate this connection explicitly, following earlier observations in [4,39],

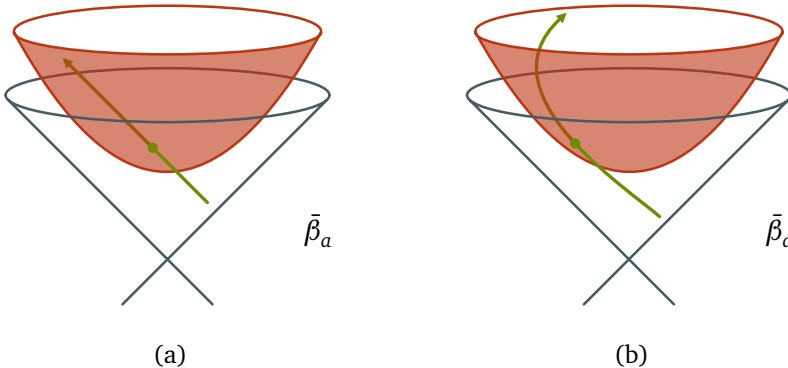

(a)            (b)

Figure 1: On shell, metrics of the form (5) map to trajectories in an external three-dimensional Minkowski spacetime. The vacuum Kasner solutions map to straight null lines, and spatial curvature or matter coupling can give curved trajectories.

This equation gives a simple constraint which we can solve for the time derivative of one of the parameters, but we can also see it as a statement on the norm of the $\beta_a$ vector. To see this more clearly, we can redefine the scaling exponents as follows,

$$\beta_1 = \frac{1}{\sqrt{6}}\left(\bar{\beta}_1 - \bar{\beta}_2 - \sqrt{3}\bar{\beta}_3\right), \tag{7a}$$

$$\beta_2 = \frac{1}{\sqrt{6}}\left(\bar{\beta}_1 - \bar{\beta}_2 + \sqrt{3}\bar{\beta}_3\right), \tag{7b}$$

$$\beta_3 = \frac{1}{\sqrt{6}}\left(\bar{\beta}_1 + 2\bar{\beta}_2\right), \tag{7c}$$

so that the equation above becomes

$$0 = \left(-\dot{\bar{\beta}}_1^2 + \dot{\bar{\beta}}_2^2 + \dot{\bar{\beta}}_3^2\right). \tag{8}$$

This means that we can think of $\dot{\bar{\beta}}_a$ as a vector in a Minkowski 'minisuperspace', and the Hamiltonian constraint encoded by the $tt$ component of the Einstein equations requires this vector to be null.

The space-time components, corresponding to the momentum constraint, are identically satisfied. The space-space components of the Einstein equations encode the evolution equations, and they imply

$$0 = \ddot{\beta}_a + \dot{\beta}_a\left(\dot{\alpha} + \dot{\beta}_1 + \dot{\beta}_2 + \dot{\beta}_3\right). \tag{9}$$

From this, we see that it is useful to redefine our choice of lapse function as follows,

$$\alpha(t) \rightarrow \alpha(t) - \left(\beta_1(t) + \beta_2(t) + \beta_3(t)\right), \tag{10}$$

so that the evolution equations become

$$0 = \dot{\alpha}\dot{\beta}_a + \ddot{\beta}_a = e^{-\alpha}\frac{d}{dt}\left(e^{\alpha}\dot{\beta}_a\right). \tag{11}$$

To further simplify this, we can then introduce a new time coordinate $\tau(t)$ such that we have $d\tau = e^{-\alpha}dt$, which absorbs what remains of the lapse function. Now using dots to denote $\tau$ derivatives, the evolution equations are then simply

$$\ddot{\beta}_a = 0, \tag{12}$$

which is the geodesic equation in flat space. Combining this with the previous equation, we see that the $\beta_a$ (or equivalently the $\bar{\beta}_a$) parametrize null geodesics in the external three-dimensional Minkowski spacetime. As a result, we can parametrize them using

$$\bar{\beta}_a(\tau) = \bar{v}_a \tau + \bar{\beta}_a^{(0)}, \tag{13}$$

where $\bar{v}_a$ is a three-dimensional null vector. By choosing the initial position $\bar{\beta}_a^{(0)}$ appropriately, we can furthermore ensure that $\bar{\beta}_a(\tau)$ lies inside the light cone of the origin of the Minkowski superspace for $\tau > 0$, as illustrated in Figure 1a above. This trajectory can then be mapped to a single point on a spacelike slice of the interior of the light cone, such as the hyperboloid in Figure 1a.

## 2.2 Mixmaster model

In the above, we saw how Kasner solutions of the vacuum Einstein equations can be reinterpreted as null geodesics in an external Minkowski spacetime parametrized by the scaling exponents. In this picture, non-trivial dynamics can arise in several ways.

First, we can modify the Kasner Ansatz (5) to include a homogeneous but anisotropic metric on spatial slices. The most famous example of this is the 'mixmaster' model [3], which we will briefly review below. Additionally, interesting dynamics can be obtained from matter couplings. In particular, we will focus on the setup proposed in [23], which obtains dynamics equivalent to that of the mixmaster model behind the horizon of an asymptotically planar AdS black hole, using three massive Abelian gauge fields. We will build on this construction to obtain the same dynamics from matter-coupled Carroll gravity in Section 4 below. For now, we give a qualitative overview of the mixmaster dynamics as it arises from its original construction which uses spatial curvature.

The idea is to replace the metric on spatial slices of the Kasner geometry (5) with a 'squashed' anisotropic version of the natural metric on $SO(3)$. To construct the latter, we parametrize a group element $g \in SO(3)$ using coordinates $x^i = (\theta, \varphi, \psi)$ as follows,

$$g(\theta, \varphi, \psi) = \exp(\psi T_3) \exp(\theta T_1) \exp(\varphi T_3). \tag{14}$$

Here, the $(T_a)_{bc} = \epsilon_{abc}$ are generators of the Lie algebra, and we set $\epsilon_{123} = +1$. In this parametrization, the Maurer–Cartan form is given by

$$\begin{aligned}
g^{-1} dg = {}& (\cos\varphi \, d\theta + \sin\varphi \sin\theta \, d\psi) T_1 + (\sin\varphi \, d\theta - \cos\varphi \sin\theta \, d\psi) T_2 \\
& + (d\varphi + \cos\theta \, d\psi) T_3.
\end{aligned} \tag{15}$$

Using the invariant Killing metric $\kappa_{ab} = \delta_{ab}$ on the Lie algebra, we obtain a homogeneous and isotropic metric on the three-dimensional $SO(3)$ group manifold,

$$d\sigma^2 = \kappa(g^{-1}dg, g^{-1}dg) = d\theta^2 + d\varphi^2 + d\psi^2 + \cos\theta \, d\phi \, d\psi. \tag{16}$$

To introduce anisotropy in this spatial metric while retaining homogeneity, we can deform the Killing metric using Kasner-type scaling exponents $\beta_a(t)$. Introducing also a lapse function $\alpha(t)$ as before, the resulting spacetime metric is

$$\begin{aligned}
ds^2 = {}& -e^{2\alpha(t)} dt^2 + e^{2\beta_1(t)}(\cos\varphi \, d\theta + \sin\varphi \sin\theta \, d\psi) \\
& + e^{2\beta_2(t)}(\sin\varphi \, d\theta - \cos\varphi \sin\theta \, d\psi) + e^{2\beta_3(t)}(d\varphi + \cos\theta \, d\psi).
\end{aligned} \tag{17}$$

The Ricci scalar of the spatial slices of this metric is given by

$$R^{(3)}(t) = e^{-2\beta_1} + e^{-2\beta_2} + e^{-2\beta_3} - \frac{1}{2}\left(e^{2(\beta_1-\beta_2-\beta_3)} + e^{2(\beta_2-\beta_3-\beta_1)} + e^{2(\beta_3-\beta_1-\beta_2)}\right), \tag{18}$$

where all scaling exponents $\beta_a(t)$ depend on time. After redefining the lapse and subsequently absorbing it by reparametrizing the time coordinate as in (10) and (11) above, the Hamiltonian constraint now gives

$$\dot{\beta}^T A \dot{\beta} = e^{2(\beta_1+\beta_2+\beta_3)} R^{(3)} = -V(\beta), \tag{19}$$

where we have introduced the potential

$$V(\beta) = \frac{1}{2}\left(e^{4\beta_1} + e^{4\beta_2} + e^{4\beta_3}\right) - \left(e^{2(\beta_1+\beta_2)} + e^{2(\beta_1+\beta_3)} + e^{2(\beta_2+\beta_3)}\right). \tag{20}$$

The evolution equations are similarly modified in terms of this potential.

We will analyze a closely related set of equations that we obtain from Carroll gravity in detail in Section 4. For now, we see that the nontrivial spatial curvature modifies the vacuum solution of Einstein's equations in terms of null geodesics in the Minkowski space of $\beta_a$ scaling exponents. Mapping these trajectories to the motion of a particle on a slicing of the interior of the light cone, as illustrated in Figure 1b, the spatial curvature acts as a potential for this particle. The first three terms of the potential dominate, and they allow the trajectories to become timelike. At late times, when the components of $\beta_a$ will typically be large, these terms lead to hard walls, restricting the trajectory to

$$\beta_1 \leq 0, \qquad \beta_2 \leq 0, \qquad \beta_3 \leq 0. \tag{21}$$

As we will see explicitly in Section 4.2, this region maps to a triangle on a spatial slice inside the light cone in $\bar{\beta}_a$ space, and the mixmaster dynamics can be described as the billiard motion of a particle on this triangular region.

## 3 Leading-order Carroll gravity

We now introduce the necessary technology for the ultra-local Carroll expansion that was developed in [53], which built on the non-relativistic Newton–Cartan expansion developed in [55–57]. As we will see, an ultra-local $c \to 0$ expansion of Lorentzian geometry results in Carroll geometry plus subleading corrections, including an appropriate notion of connection and curvature. This ultra-local structure is naturally adapted to the near-singularity region of black holes, as illustrated in Figure 2 below.

Applied to the Einstein–Hilbert action, the leading-order theory of Carroll gravity obtained from this ultra-local expansion is given by [53][2]

$$\frac{c^3}{2\kappa} \int d^d x \sqrt{-g}\, R = \frac{c^2}{2\kappa} \int d^d x\, e\left(K^{\mu\nu} K_{\mu\nu} - K^2\right) + \mathcal{O}(c^0), \tag{22}$$

where $K_{\mu\nu}$ is the extrinsic curvature of spatial slices, as we will see below. Note that this is precisely the kinetic part of the ADM Lagrangian in fully covariant notation. Likewise, applying the same expansion procedure to the Maxwell action leads to

$$-\frac{1}{4gc} \int d^d x \sqrt{-g}\, g^{\mu\nu} g^{\rho\sigma} F_{\mu\rho} F_{\nu\sigma} = \frac{1}{2gc^2} \int d^d x\, e\, h^{\mu\nu} E_\mu E_\nu + \mathcal{O}(c^0). \tag{23}$$

As we will see, the tensor $h^{\mu\nu}$ can be interpreted as the inverse of a (degenerate) spatial metric $h_{\mu\nu}$ which, together with a 'time' vector field $v^\mu$ determines the leading-order Carroll geometry.[3] Since the leading-order Carroll limit of the Maxwell action only contains the electric

---

[2]This equality holds up to boundary terms. The resulting action also appeared in [50–52].

[3]Since the spatial metric $h_{\mu\nu}$ is degenerate, the inverse $h^{\mu\nu}$ is not uniquely determined, which is reflected by its transformation (28) under local Carroll boosts.

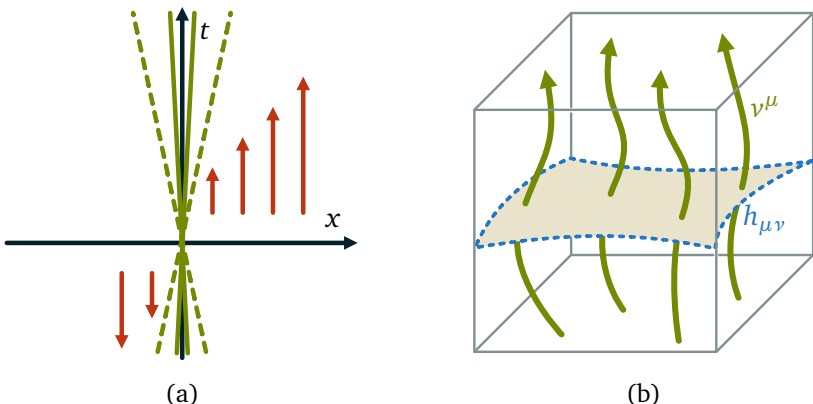

Figure 2: Illustration of (a) the ultra-local Carroll limit of the Lorentzian causal structure, where light cones are contracted to a line, together with its subleading corrections, and (b) the curved Carroll geometry with time vector field $v^\mu$ and spatial degenerate metric $h_{\mu\nu}$, which appears naturally near spacelike singularities.

field, it is often called the 'electric' limit of electromagnetism [50, 52, 60–62]. By analogy, the leading-order Carroll limit (22) obtained from the Einstein–Hilbert action is also sometimes known as the 'electric Carroll limit' of general relativity.[4]

As in ordinary Lorentzian gravity, actions for Carroll matter can be coupled to Carroll gravity, which results in metric equations of motion sourced by the corresponding energy-momentum tensors. By projecting these equations along components tangential and normal to a spatial hypersurface, we obtain a decomposition in constraint and evolution equations. Due to their ultra-locality, the evolution equation of these models is particularly simple, and we will discuss some of its general features in Section 3.2. We then work out the specific example of 'electric' Carroll gravity (22) coupled to the electric limit (23) of the Maxwell action in Section 3.3.

## 3.1 Some background on Carroll geometry

Our approach to the ultra-local $c \to 0$ Carroll limit and its subleading corrections [53] starts by explicitly introducing a speed of light in our Lorentzian spacetime metric,

$$g_{\mu\nu} = c^2 T_\mu T_\nu + \Pi_{\mu\nu}. \tag{24}$$

This decomposition singles out a particular timelike vielbein $T_\mu$, in analogy to the choice of spatial slicing implicit in the Ansatz (5). The timelike vielbein $T_\mu$ and the spatial vielbeine contained in $\Pi_{\mu\nu}$ form a local frame in a given Lorentzian geometry, around which we define our ultra-local expansion. This frame can be complemented with a corresponding decomposition of the inverse metric,

$$g^{\mu\nu} = -\frac{1}{c^2} V^\mu V^\nu + \Pi^{\mu\nu}. \tag{25}$$

where $V^\mu$ is the inverse timelike vielbein and $\Pi^{\mu\nu}$ contains the inverse spacelike vielbeine. As such, they satisfy the usual orthonormality and completeness relations

$$T_\mu V^\mu = -1, \qquad T_\mu \Pi^{\mu\nu} = 0, \qquad V^\mu \Pi_{\mu\nu} = 0, \qquad -V^\mu T_\mu + \Pi^{\mu\rho}\Pi_{\rho\nu} = \delta^\mu_\nu. \tag{26}$$

---

[4]Using different parametrizations, 'magnetic' Carroll limits can also be obtained, but we will not need them here. Whereas electric actions typically only contain time derivatives of the fields (corresponding to the extrinsic curvature in gravity), magnetic Carroll actions typically contain only spatial derivatives (or spatial curvature in gravity). To retain Carroll invariance, the latter also come with constraints that strongly restrict their dynamics in time, so they are unsuitable for describing BKL dynamics.

If the one-form $T_\mu$ satisfies the integrability condition $T \wedge dT = 0$, we can think of it as defining a spatial hypersurface, and $\Pi_{\mu\nu}$ then is the induced metric on this hypersurface. In Lorentzian geometry, a decomposition such as (24) is not invariant under local Lorentz boosts, which mix spacelike and timelike vielbeine. As we will see shortly, this carries over to the Carroll boosts that arise from such local Lorentz boosts. Nonetheless, even though they are not boost-invariant, it will often be useful to use 'spacelike' and 'timelike' projections for our Carroll equations, in particular when solving equations of motion.

After implementing this decomposition, we expand its geometric variables as follows,

$$V^\mu = v^\mu + c^2 M^\mu + \mathcal{O}(c^4), \qquad \Pi^{\mu\nu} = h^{\mu\nu} + c^2 \Phi^{\mu\nu} + \mathcal{O}(c^4), \tag{27a}$$
$$T_\mu = \tau_\mu + \mathcal{O}(c^2), \qquad \Pi_{\mu\nu} = h_{\mu\nu} + \mathcal{O}(c^2), \tag{27b}$$

where the subleading terms in the second line can be obtained from those in the first line using the expansion of (26). In this expansion, we assume that our metric variables are analytic, and we furthermore only take even powers into account. This truncation is self-consistent, and it will furthermore prove to be sufficient for our purposes.

At leading order, the local Lorentz boosts of our original Lorentz frame result in

$$\delta v^\mu = 0, \qquad \delta \tau_\mu = \lambda_\mu, \qquad \delta h^{\mu\nu} = v^\mu \lambda^\nu + \lambda^\mu v^\nu, \qquad \delta h_{\mu\nu} = 0, \tag{28}$$

which are known as Carroll boosts. The parameter $\lambda_\mu$ is spatial in the sense that it satisfies $v^\mu \lambda_\mu = 0$, which means that it can be consistently raised as $\lambda^\mu = h^{\mu\rho} \lambda_\rho$ without this being ambiguous under the boost transformations itself.

We see that the time vector field $v^\mu$ and the spatial metric $h_{\mu\nu}$ are invariant under Carroll boosts. As such, they can be seen as the fundamental metric-like quantities of Carroll geometry, as illustrated in Figure 2b. Other boost-invariant quantities include the vielbein determinant $e = \det(\tau_\mu \tau_\nu + h_{\mu\nu})$ as well as

$$K_{\mu\nu} = -\frac{1}{2} \mathcal{L}_v h_{\mu\nu}, \tag{29}$$

which we refer to as the extrinsic curvature of spatial slices. It is easy to see that $v^\mu K_{\mu\nu} = 0$, which implies that contractions such as $K = h^{\mu\nu} K_{\mu\nu}$ are also boost-invariant.

In this geometric Carroll expansion, it is furthermore convenient to replace the Levi-Civita connection of the Lorentzian geometry with a connection $\tilde{\nabla}$ that is compatible with the boost-invariant Carroll metric variables $v^\mu$ and $h_{\mu\nu}$,

$$\tilde{\nabla}_\rho v^\mu = 0, \qquad \tilde{\nabla}_\rho h_{\mu\nu} = 0. \tag{30}$$

Several choices of such connections are possible, but they all have nonzero 'intrinsic' torsion on generic backgrounds [63]. In the following, the only property we will need of our Carroll connection of choice is that it projects to a (Euclidean) Levi-Civita connection on spatial hypersurfaces [53]. This means that fully-projected derivatives such as

$$h^{\mu\rho} h^{\nu\sigma} \tilde{\nabla}_\rho \omega_\sigma, \tag{31}$$

of spatial tensors (which satisfy $v^\mu \omega_\mu = 0$) can be computed in terms of the three-dimensional Levi-Civita covariant derivative on spatial slices.

## 3.2 Equations of motion with general matter coupling

Applying the metric decomposition (24) to the Einstein–Hilbert action, we obtain [53]

$$\frac{c^3}{2\kappa} \int d^d x \sqrt{-g} R = \frac{c^2}{2\kappa} \int d^d x \, e \left[ \left( \mathcal{K}^{\mu\nu} \mathcal{K}_{\mu\nu} - \mathcal{K}^2 \right) + c^2 \, \Pi^{\mu\nu} \overset{\scriptscriptstyle(c)}{R}_{\mu\nu} + c^4 \, \Pi^{\mu\rho} \Pi^{\nu\sigma} \partial_{[\mu} T_{\nu]} \partial_{[\rho} T_{\sigma]} \right]. \tag{32}$$

This is known as the 'pre-ultra-local' parametrization of the Einstein–Hilbert action, as it still describes the full dynamics of Einstein gravity, but using variables that are adapted to the ultra-local $c \to 0$ expansion. To leading order in this expansion, the Ricci tensor and extrinsic curvature terms in (32) reduce to the Carroll quantities introduced in the previous section,

$$\mathcal{K}_{\mu\nu} = -\frac{1}{2}\mathcal{L}_V \Pi_{\mu\nu} = K_{\mu\nu} + \mathcal{O}(c^2), \tag{33}$$

$$\overset{\scriptscriptstyle(c)}{R}_{\mu\nu} = \tilde{R}_{\mu\nu} + \mathcal{O}(c^2). \tag{34}$$

For our present purposes, we will not need the subleading order terms in such expansions. Indeed, we will only need the leading-order terms in the expansion of the action (32),

$$\frac{c^3}{2\kappa}\int d^d x \sqrt{-g}\, R = \frac{c^2}{2\kappa}\int d^d x\, e\left(K^{\mu\nu}K_{\mu\nu} - K^2\right) + \mathcal{O}(c^0). \tag{35}$$

As we mentioned above, up to an overall factor of $c^2$, this gives what is usually referred to as the 'electric' Carroll limit of Einstein gravity. Note that all Riemann curvature terms in (32) only appear from subleading order onwards, and they are thus suppressed in this leading-order electric action.

The same procedure can be applied to obtain an 'electric' Carroll matter Lagrangian from a given field theory coupled to Lorentzian geometry. The result is a field theory with Carroll background geometry, which therefore can be coupled to dynamical Carroll gravity. Given such an action for matter-coupled Carroll gravity,

$$S_e[v, h] + S_{\mathrm{m}}[\phi; v, h] = \frac{1}{2\kappa}\int d^d x\, e\left(K^{\mu\nu}K_{\mu\nu} - K^2\right) + S_{\mathrm{m}}[\phi; v, h], \tag{36}$$

we first write down the metric equations of motion and then consider the specific example of abelian Yang–Mills (23) which we will treat in detail below.

Varying the gravity action with respect to $v^\mu$ and $h^{\mu\nu}$ gives

$$\delta S_e = \frac{1}{2\kappa}\int d^d x\, e\left[2G^\nu_\mu \delta v^\mu + G^h_{\mu\nu}\delta h^{\mu\nu}\right], \tag{37}$$

where we have [53]

$$G^\nu_\mu = -\frac{1}{2}\tau_\mu\left(K^{\rho\sigma}K_{\rho\sigma} - K^2\right) - h^{\nu\rho}\tilde{\nabla}_\rho\left(K_{\mu\nu} - Kh_{\mu\nu}\right), \tag{38a}$$

$$G^h_{\mu\nu} = -\frac{1}{2}h_{\mu\nu}\left(K^{\rho\sigma}K_{\rho\sigma} - K^2\right) + K\left(K_{\mu\nu} - Kh_{\mu\nu}\right) - v^\rho\tilde{\nabla}_\rho\left(K_{\mu\nu} - Kh_{\mu\nu}\right). \tag{38b}$$

Varying the matter action with respect to the background metric gives the currents

$$\delta S_m = -\int d^d x\, e\left(T^\nu_\mu \delta v^\mu + \frac{1}{2}T^h_{\mu\nu}\delta h^{\mu\nu}\right). \tag{39}$$

They can be combined into a total boost-invariant energy-momentum tensor [64–66]

$$T^\mu{}_\nu = -v^\mu T^\nu_\nu - h^{\mu\rho}T^h_{\rho\nu}, \tag{40}$$

but we only need the individual currents for the gravity equations of motion.

Combining the variations (37) and (39) we get the following matter-coupled equations of motion

$$G^\nu_\mu = \kappa\, T^\nu_\mu, \qquad G^h_{\mu\nu} = \kappa\, T^h_{\mu\nu}. \tag{41}$$

It is useful to take the spatial and temporal projections of these equations of motion using the projectors $-v^\mu \tau_\mu$ and $h^{\mu\rho}h_{\rho\nu}$, which leads to [53, 60],

$$\frac{1}{2}\left(K^{\rho\sigma}K_{\rho\sigma} - K^2\right) = v^\mu G_\mu^\nu = \kappa\, v^\mu T_\mu^\nu\,, \tag{42a}$$

$$-h^{\alpha\mu}h^{\rho\nu}\tilde{\nabla}_\rho\left(K_{\mu\nu} - Kh_{\mu\nu}\right) = h^{\alpha\mu}G_\mu^\nu = \kappa\, h^{\alpha\mu}T_\mu^\nu\,, \tag{42b}$$

$$\mathcal{L}_\nu K_{\mu\nu} = KK_{\mu\nu} - 2K_\mu^{\ \rho}K_{\rho\nu} - \kappa\, h_\mu^\alpha h_\nu^\beta T_{\alpha\beta}^h + \frac{\kappa}{(d-1)}h_{\mu\nu}\left[T_\rho^\nu v^\rho + T_{\rho\sigma}^h h^{\rho\sigma}\right]. \tag{42c}$$

We can interpret the first two equations as constraints on initial data $(h_{\mu\nu}, K_{\mu\nu})$ on a given initial time slice. The third equation the determines the evolution of this initial data along a given $v^\mu$ time vector. In fact, the equations above can be identified with subsets of the full (covariant) ADM constraint and evolution equations for Einstein gravity, where particular terms, such as the three-dimensional Ricci scalar, drop out in the Carroll limit. As emphasized in [53], the fact that these terms are suppressed makes the equations significantly simpler than their full Lorentzian analogues in general relativity. In particular, instead of the usual hyperbolic partial differential equation, the evolution equation is just an ordinary differential equation with respect to the time coordinate associated to the $v^\mu$ vector field. As we will see explicitly in Section 4.2, this ultra-local simplification makes the equations easily tractable.

## 3.3 Carroll gravity coupled to electric limit of Maxwell

Now let us consider the specific example of the Carroll limit of gravity coupled to electromagnetism. Following a similar procedure as what we outlined above for the Einstein–Hilbert action, the leading-order term in the ultra-local $c \to 0$ limit of the Maxwell action leads to the 'electric' Carroll limit of electromagnetism [50, 60, 61]

$$S_{\text{EMe}} = \frac{1}{2g}\int d^d x\, e\, v^\mu v^\nu h^{\rho\sigma}F_{\mu\rho}F_{\nu\sigma}\,, \tag{43}$$

where we identify $E_\mu = v^\rho F_{\rho\nu}$ with the electric field. Varying the action with respect to the gauge field, we get the matter equation of motion

$$0 = \partial_\mu\left(e\, v^{[\mu}h^{\nu]\rho}v^\sigma F_{\rho\sigma}\right). \tag{44}$$

Next, varying the action with respect to $v^\mu$ and $h^{\mu\nu}$ as in (39) leads to the energy-momentum currents

$$T_\mu^\nu = -\frac{1}{2g}\left[\tau_\mu\left(v^\alpha F_{\alpha\rho}h^{\rho\sigma}v^\beta F_{\beta\sigma}\right) + 2F_{\mu\rho}h^{\rho\sigma}v^\beta F_{\beta\sigma}\right], \tag{45a}$$

$$T_{\mu\nu}^h = \frac{1}{2g}\left[h_{\mu\nu}\left(v^\alpha F_{\alpha\rho}h^{\rho\sigma}v^\beta F_{\beta\sigma}\right) - 2v^\alpha F_{\alpha\mu}v^\beta F_{\beta\nu}\right]. \tag{45b}$$

Note that we have $v^\mu h^{\nu\rho}T_{\mu\nu}^h = 0$, which corresponds to a Ward identity due to Carroll boost invariance. Additionally, note that the last term in $T_\mu^\nu$ contains a contribution proportional to $h^{\alpha\mu}F_{\mu\rho}h^{\rho\sigma}$, which encodes the magnetic field. Even though the action (43) only depends on the electric field, this term arises from the variation of the $v^\mu$ fields. For our purposes, we will only need the electric field coupling, so we will set the magnetic field to zero in the following.

With this, the Hamiltonian constraint (42a), the momentum constraint (42b) and the evolution equation (42c) resulting from the coupling of leading-order Carroll gravity to the electric Carroll limit of the Maxwell action (43) are given by [60]

$$\frac{1}{2}\left(K^{\rho\sigma}K_{\rho\sigma} - K^2\right) = -\frac{\kappa}{2g}h^{\mu\nu}E_\mu E_\nu, \tag{46a}$$

$$-h^{\alpha\mu}h^{\rho\nu}\tilde{\nabla}_\rho\left(K_{\mu\nu} - Kh_{\mu\nu}\right) = 0, \tag{46b}$$

$$\mathcal{L}_\nu K_{\mu\nu} - KK_{\mu\nu} + 2K_\mu{}^\rho K_{\rho\nu} = \frac{\kappa}{g}\left(E_\mu E_\nu - \frac{h_{\mu\nu}}{(d-1)}E^\rho E_\rho\right). \tag{46c}$$

In the following, we will consider a straightforward generalization of these equations, involving not one but three gauge fields.

# 4 Mixmaster dynamics from Carroll gravity

We now want to show that the Carroll theories of gravity obtained from an ultra-local expansion of general relativity can capture BKL dynamics. Mirroring the metric Ansatz (5) that we used in Einstein gravity, we now take the Carroll Ansatz

$$v^\mu\partial_\mu = -e^{\alpha(t)}\partial_t, \qquad h_{\mu\nu}dx^\mu dx^\nu = e^{2\beta_x(t)}dx^2 + e^{2\beta_y(t)}dy^2 + e^{2\beta_z(t)}dz^2. \tag{47}$$

In Section 4.1, we first show that evaluating equations of motion of pure Carroll gravity on this Ansatz leads to equations equivalent to the ones we obtained from pure Einstein gravity in Section 2.1. As before, this leads to the interpretation of the vacuum Kasner-type solutions as null geodesics in a three-dimensional Minkowski minisuperspace parametrized by the scaling exponents. Next, inspired by the bottom-up AdS/CFT model recently proposed in [23], we consider Carroll gravity coupled to three electric gauge fields in Section 4.2. We show that the resulting equations of motion fully capture the dynamics of the mixmaster model [67] that we briefly discussed in Section 2.2.

To begin, let us list some further geometric properties of our Ansatz (47). While the inverse spatial metric $h^{\mu\nu}$ transforms nontrivially under the local Carroll boosts (28), which affects its space-time components, its space-space components are unambiguous. Using the coordinates $x^\mu = (t, x^i)$ from the Ansatz above, where $x^i = (x, y, z)$ are the spatial coordinates, we have

$$h^{ij}\partial_i\partial_j = e^{-2\beta_x(t)}\partial_x^2 + e^{-2\beta_y(t)}\partial_y^2 + e^{-2\beta_z(t)}\partial_z^2. \tag{48}$$

The square root of the metric determinant, the extrinsic curvature and its trace are

$$e = \sqrt{\det\left(\tau_\mu\tau_\nu + h_{\mu\nu}\right)} = e^{-\alpha+\beta_x+\beta_y+\beta_z} = e^{-\alpha}\sqrt{h}, \tag{49a}$$

$$K_{\mu\nu}dx^\mu dx^\nu = e^\alpha(\dot{\beta}_x e^{2\beta_x}dx^2 + \dot{\beta}_y e^{2\beta_y}dy^2 + \dot{\beta}_z e^{2\beta_z}dz^2), \tag{49b}$$

$$K = h^{ij}K_{ij} = e^\alpha\left(\dot{\beta}_x + \dot{\beta}_y + \dot{\beta}_z\right). \tag{49c}$$

As we mentioned previously, the extrinsic curvature is purely spatial. For this reason, contractions such as the trace (49c) are only sensitive to the spatial components (4.2) of the inverse spatial metric, which are not modified by local Carroll boosts.

### 4.1 Kasner geometries in vacuum

In vacuum, the equations of motion (42) for leading-order Carroll gravity reduce to

$$0 = \frac{1}{2}\left(K^{\rho\sigma}K_{\rho\sigma} - K^2\right), \tag{50a}$$

$$0 = -h^{\alpha\mu}h^{\rho\nu}\tilde{\nabla}_\rho\left(K_{\mu\nu} - Kh_{\mu\nu}\right), \tag{50b}$$

$$\mathcal{L}_\nu K_{\mu\nu} = KK_{\mu\nu} - 2K_\mu{}^\rho K_{\rho\nu}. \tag{50c}$$

Note that the Carroll covariant derivative $\tilde{\nabla}_\rho$ in the momentum constraint (50b) is fully projected onto the spatial hypersurface, where it simply reduces to the Levi-Civita covariant derivative, as we discussed around (31) above. As a result, using the Ansatz (47) the right-hand side of (50b) will vanish identically, since the extrinsic curvature $K_{\mu\nu}$, its trace $K$ and the spatial metric $h_{\mu\nu}$ are independent of the spatial coordinates. Therefore, the momentum constraint is identically satisfied.

The Hamiltonian constraint gives

$$0 = \frac{1}{2}\left(K^{ij}K_{ij} - K^2\right) = \frac{1}{2}e^{2\alpha}\left[\left(\dot{\beta}_1^2 + \dot{\beta}_2^2 + \dot{\beta}_3^2\right) - \left(\dot{\beta}_1 + \dot{\beta}_2 + \dot{\beta}_3\right)^2\right] \tag{51}$$

$$= e^{2\alpha}\left(-\dot{\bar{\beta}}_1^2 + \dot{\bar{\beta}}_2^2 + \dot{\bar{\beta}}_3^2\right). \tag{52}$$

This is equivalent to what we got from the time-time component of the Einstein equation in (6) above. Using the transformation (7), we see that $\dot{\bar{\beta}}_a$ is a null vector in a three-dimensional Minkowski minisuperspace. The evolution equation (50c) then leads to

$$0 = e^{2(\alpha+\beta_a)}\left[\ddot{\beta}_a + \dot{\beta}_a\left(\dot{\alpha} + \dot{\beta}_1 + \dot{\beta}_2 + \dot{\beta}_3\right)\right]. \tag{53}$$

Up to an overall prefactor, this is what we obtained from the Einstein equations in (9). As we did there, it is useful to first shift the lapse function $\alpha(t)$ as in (10), and subsequently reparametrize $t = t(\tau)$ to absorb the lapse completely. The evolution equations then reduce to

$$\ddot{\bar{\beta}}_a = 0. \tag{54}$$

Together with the constraint equation (51), this means that the trajectories parametrized by $\beta_a(\tau)$ (or equivalently by $\bar{\beta}_a(\tau)$) correspond to null geodesics, as in Figure 1a.

### 4.2 Mixmaster dynamics from electric matter coupling

We now consider coupling Carroll geometries of the form (47) to three copies of the electric Carroll gauge field described by the action (43). For the gauge fields, we take

$$A^1 = f_1(t)dx, \qquad A^2 = f_2(t)dy, \qquad A^3 = f_3(t)dz. \tag{55}$$

Solving the gauge field equations of motion (44) on the background (47), we obtain the following solutions, with $\phi_a$ arbitrary constants,

$$\dot{f}_1 = \phi_1 e^{-\alpha}e^{\beta_1-\beta_2-\beta_3}, \quad \dot{f}_2 = \phi_2 e^{-\alpha}e^{\beta_2-\beta_3-\beta_1}, \quad \dot{f}_3 = \phi_3 e^{-\alpha}e^{\beta_3-\beta_1-\beta_2}. \tag{56}$$

This leads to electric fields $E^a_\mu = v^\rho F^a_{\rho\nu}$ along each of the spatial axes, given by

$$E^1 = -\phi_x e^{\beta_x-\beta_y-\beta_z}dx, \quad E^2 = -\phi_y e^{\beta_y-\beta_z-\beta_x}dx, \quad E^3 = -\phi_z e^{\beta_z-\beta_x-\beta_y}dx, \tag{57}$$

and the magnetic fields $h^{\mu\rho}h^{\nu\sigma}F_{\rho\sigma}$ vanish identically. With this matter content, the constraint and evolution equations (46) give

$$\frac{1}{2}\left(K^{\rho\sigma}K_{\rho\sigma}-K^2\right) = -\frac{\kappa}{2g}h^{\mu\nu}\left(E^1_\mu E^1_\nu + E^2_\mu E^2_\nu + E^3_\mu E^3_\nu\right), \tag{58a}$$

$$-h^{\alpha\mu}h^{\rho\nu}\tilde{\nabla}_\rho\left(K_{\mu\nu}-Kh_{\mu\nu}\right)=0, \tag{58b}$$

$$\mathcal{L}_\nu K_{\mu\nu}-KK_{\mu\nu}+2K_\mu{}^\rho K_{\rho\nu} = \frac{\kappa}{g}\left(E^1_\mu E^1_\nu + E^2_\mu E^2_\nu + E^3_\mu E^3_\nu\right) \tag{58c}$$

$$-\frac{\kappa}{g}\frac{h_{\mu\nu}}{(d-1)}h^{\rho\sigma}\left(E^1_\rho E^1_\sigma + E^2_\rho E^2_\sigma + E^3_\rho E^3_\sigma\right).$$

Note that there are no source terms in the momentum equation, which is therefore again identically satisfied since there is no spatial dependence in the metric Ansatz (47).

Again, it is now convenient to redefine the lapse and subsequently absorb it by reparametrizing the time coordinate $t = t(\tau)$ such that

$$\alpha(t)\to\alpha(t)-\left(\beta_x+\beta_y+\beta_z\right), \qquad e^{\alpha(t)}\frac{d}{dt}\beta(t)=\frac{d}{d\tau}\beta(t(\tau)). \tag{59}$$

The Hamiltonian constraint (58a) and the evolution equations (58c) then become

$$-2\left(\dot{\beta}_1\dot{\beta}_2+\dot{\beta}_1\dot{\beta}_3+\dot{\beta}_2\dot{\beta}_3\right)=-\frac{\kappa}{g}V(\beta), \qquad \ddot{\beta}_a=\frac{\kappa}{2g}\left(1-\partial_{\beta_a}\right)V(\beta), \tag{60}$$

where the following potential is sourced by the gauge field couplings,

$$V(\beta)=\phi_1^2 e^{2\beta_1}+\phi_2^2 e^{2\beta_2}+\phi_3^2 e^{2\beta_3}. \tag{61}$$

In regions where this potential and its derivatives are negligible, we see that we recover the vacuum solutions discussed in the previous section. In particular, in such regions, the trajectory $\beta_a(\tau)$ is approximately given by the straight null lines

$$\beta_a(\tau)\approx v_a\tau+\beta_a^{(0)}. \tag{62}$$

Additionally, we see that as $\beta_a$ grows in time, the potential (61) will become exponentially steep. At late times, the potential therefore effectively bounds off the region

$$\beta_1\leq 0, \qquad \beta_2\leq 0, \qquad \beta_3\leq 0, \tag{63}$$

as in Equation (21) for the $SO(3)$ mixmaster model. When the null trajectories reach the boundary of this region, the potential peaks, allowing the tangent vector $\beta_a(\tau)$ to briefly become timelike, bouncing the trajectory off the potential wall and returning to another approximately null trajectory.

To further analyze the resulting dynamics, it is useful to change variables to the barred coordinates introduced in (7). The constraint equation then becomes

$$-\dot{\bar{\beta}}_1^2+\dot{\bar{\beta}}_2^2+\dot{\bar{\beta}}_3^2=-\frac{\kappa}{g}V\left(\beta(\bar{\beta})\right) \tag{64}$$

$$=-\frac{\kappa}{g}\left(\phi_1^2 e^{\sqrt{2/3}(\bar{\beta}_1-\bar{\beta}_2-\sqrt{3}\bar{\beta}_3)}+\phi_2^2 e^{\sqrt{2/3}(\bar{\beta}_1-\bar{\beta}_2+\sqrt{3}\bar{\beta}_3)}+\phi_3^2 e^{\sqrt{2/3}(\bar{\beta}_1+2\bar{\beta}_2)}\right),$$

while the evolution equations become

$$\ddot{\bar{\beta}}_1=\frac{1}{\sqrt{6}}\frac{\kappa}{g}\left[\phi_1^2 e^{\sqrt{2/3}(\bar{\beta}_1-\bar{\beta}_2-\sqrt{3}\bar{\beta}_3)}+\phi_2^2 e^{\sqrt{2/3}(\bar{\beta}_1-\bar{\beta}_2+\sqrt{3}\bar{\beta}_3)}+\phi_3^2 e^{\sqrt{2/3}(\bar{\beta}_1+2\bar{\beta}_2)}\right], \tag{65a}$$

$$\ddot{\bar{\beta}}_2=\frac{1}{\sqrt{6}}\frac{\kappa}{g}\left[\phi_1^2 e^{\sqrt{2/3}(\bar{\beta}_1-\bar{\beta}_2-\sqrt{3}\bar{\beta}_3)}+\phi_2^2 e^{\sqrt{2/3}(\bar{\beta}_1-\bar{\beta}_2+\sqrt{3}\bar{\beta}_3)}-2\phi_3^2 e^{\sqrt{2/3}(\bar{\beta}_1+2\bar{\beta}_2)}\right], \tag{65b}$$

$$\ddot{\bar{\beta}}_3=\frac{1}{\sqrt{2}}\frac{\kappa}{g}\left[\phi_1^2 e^{\sqrt{2/3}(\bar{\beta}_1-\bar{\beta}_2-\sqrt{3}\bar{\beta}_3)}-\phi_2^2 e^{\sqrt{2/3}(\bar{\beta}_1-\bar{\beta}_2+\sqrt{3}\bar{\beta}_3)}\right]. \tag{65c}$$

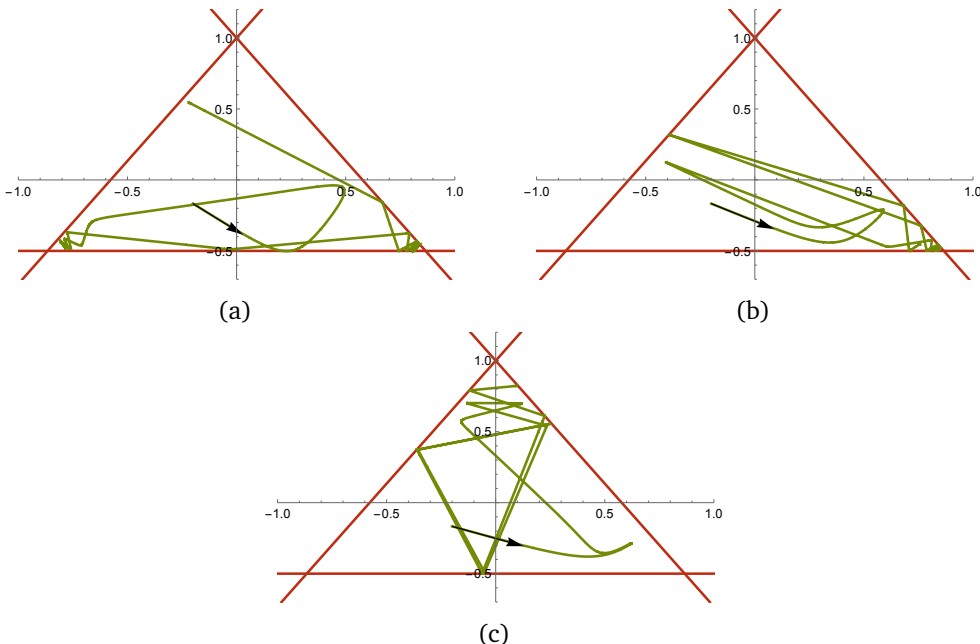

Figure 3: Three sample evolutions of the equations (65) in the $\gamma_a$ variables (67) from $\tau = 0$ to $\log(1 + \tau) = 23$, showing strong dependence on the initial conditions. All trajectories start from $\gamma_1 = -1/5$, $\gamma_2 = -1/6$ and $\gamma_3 = 0$. We solve the initial velocity $\dot{\gamma}_1$ from the constraint (64) with $\dot{\gamma}_3 = 2$ and (a) $\dot{\gamma}_2 = -11/10$, which gives a trajectory that bounces around in the lower left and right corner before flying off towards the top corner, while taking (b) $\dot{\gamma}_2 = -9/10$, it bounces between the lower right corner and the upper left wall, and with (c) $\dot{\gamma}_2 = -7/10$, the trajectory bounces between all three walls.

In these coordinates, the potential walls (63) parametrize a cone with triangular cross-sections for fixed (negative) values of $\bar{\beta}_1$,

$$\bar{\beta}_2 \leq -\bar{\beta}_1/2, \quad \bar{\beta}_2 \geq \bar{\beta}_1 - \sqrt{3}\bar{\beta}_3, \quad \bar{\beta}_2 \geq \bar{\beta}_1 + \sqrt{3}\bar{\beta}_3. \tag{66}$$

Using hyperbolic slices, this would map to a triangle in hyperbolic space. However, we will just use a flat slicing for simplicity. Correspondingly, we introduce coordinates

$$\gamma_1 = \frac{\bar{\beta}_2}{\bar{\beta}_1}, \qquad \gamma_2 = \frac{\bar{\beta}_3}{\bar{\beta}_1}, \qquad \gamma_3 = \log(-\bar{\beta}_1), \tag{67}$$

such that the region bounded by the potential is always given by

$$\gamma_1 \geq -1/2, \quad \gamma_1 \leq 1 - \sqrt{3}\gamma_2, \quad \gamma_1 \leq 1 + \sqrt{3}\gamma_2. \tag{68}$$

We plot a few examples of the resulting dynamics in Figure 3 below. As time progresses, the deflection of the trajectories becomes increasingly sharp, ending up with billiard dynamics inside the triangular region (68).

## 5 Summary and outlook

In this work, we have put forward the ultra-local Carroll expansion of general relativity as a novel and useful tool for studying Belinski–Khalatnikov–Lifshitz (BKL) dynamics near spacelike

singularities. As a first example, following earlier observations in [4, 39], we have explicitly shown that mixmaster dynamics can be obtained from the leading-order Carroll limit of gravity coupled to three abelian gauge fields, following the bottom-up AdS/CFT model which was introduced recently in [23]. This further suggests that Carroll expansions may be a useful tool in constructing new tractable models of the dynamics beyond the horizon in holography.

Several immediate followup studies are possible. First, since the leading-order Carroll gravity models we considered here implement the strict ultra-local limit off shell, the resulting evolution equations (42c) are ordinary differential equations for *all* choices of initial data. In the above, we have used a spatially homogeneous Carroll metric Ansatz to mirror the minisuperspace models in Einstein gravity that we aimed to reproduce. However, we can in fact also easily obtain tractable models incorporating spatial inhomogeneity, in contrast to Einstein gravity, where spatial inhomogeneity almost always leads to partial differential equations and greatly complicates solving the evolution equations.

While BKL dynamics is often argued to be generic in the late time limit for any choice of initial data, the formidable complexity of the full evolution equations makes it hard to explicitly see the emergence of BKL from inhomogeneous initial data without resorting to intricate numerical simulations. It would be very interesting to see if our Carroll models can reproduce key features that are observed in such simulations of Einstein gravity, such as late-time spikes [68], in a much more straightforward numerical setting, and we hope to return to this shortly. Additionally, in the context of AdS/CMT, it could be interesting to investigate if these phenomena may have some boundary interpretation in well-known setups breaking translation symmetry in for example holographic superconductors.

Next, the Carroll gravity models we employed above are in fact only the leading order theories in a systematic ultra-local expansion of general relativity [53]. What is more, the intrinsic curvature of spatial slices does not enter into this expansion until next-to-leading order. As we illustrated in Section 2.2, spatial curvature plays a key role in BKL dynamics by sourcing potential walls such as in the original $SO(3)$ or Bianchi IX mixmaster model [3]. In the above, building on earlier observations in [4, 39], we were able to reproduce dynamics equivalent to the mixmaster model in a Carroll limit of the model proposed in [23], using only potential walls sourced by matter. However, to capture the full richness of BKL dynamics, our current Carroll models should be extended to also be sensitive to spatial curvature and its resulting potential walls.

It would also be very interesting to use subleading orders in the Carroll expansion of general relativity to obtain an analytic or numerical description of the subleading corrections to BKL limits. One can furthermore hope that such subleading corrections may help to extend our understanding of BKL dynamics in holography further away from the singularity and towards the horizon. Specifically, the bulk Carroll dynamics that we studied appears at late interior times, which begs the question if it has an interpretation in terms of a boundary RG flow. Likewise, it would be interesting to see if the Carroll limit has an imprint in the question of identifying boundary observables that reconstruct the experience of an infalling observer in the bulk [69, 70].

# Acknowledgments

We are thankful to Ján Pulmann and Dmitriy Zhigunov for useful discussions, and to Jelle Hartong and Marc Henneaux for useful discussions and helpful comments on an earlier version of this paper. GO would like to thank the organizers and participants of the 'Eurostrings 2024 meets Fundamental Physics UK' conference (Southampton) and the 2024 'Carrollian Physics and Holography' workshop (Erwin Schrödinger Institute, Vienna), where part of this work was presented.



**Funding information**   The work of GO is supported by the Royal Society URF of Jelle Hartong through the Research Fellows Enhanced Research Expenses 2022 (RF\ERE\221013). JFP is supported by the 'Atracción de Talento' program (Comunidad de Madrid) grant 2020-T1/TIC-20495, by the Spanish Research Agency via grants CEX2020-001007-S and PID2021-123017NB-I00, funded by MCIN/AEI/10.13039/501100011033, and by ERDF 'A way of making Europe.'

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
