# Peer review of "Mixmasters in Wonderland: Chaotic dynamics from Carroll limits of gravity"

_SciPost Physics, doi:SciPost Phys. Core 8, 025 (2025)_

## Round 2 · Referee Report · Anonymous (Referee 1) · 2024-12-17

Strengths

  1. The paper shows an interesting connection between Carroll gravity and Kasner/mixmaster models used to study dynamics close to curvature singularities. The result for the mixmaster model is new and original.
  2. Clear exposition, clear calculations.
  3. Potential new applications to study near-singularity physics in AdS/CFT from the boundary by using Carroll symmetry.

Weaknesses

  1. The progress is good, but somewhat limited. Sections 2 and 3 are essentially reviews, and section 4 contains some new results. For the Kasner geometries, this was long expected (but shown explicitly here), and for the mixmaster model, it is an honest new result.
  2. The consequences of having a Carroll description of the mixmaster model remain a bit unclear. What did we now learn from this about BKL dynamics now? Some comments addressing this are in the outlook.

Report

I find the paper of good quality and with enough new material to recommend publication.

Requested changes

I think formula (2.13) is not quite correct. Are there squares missing over these one-forms?

Recommendation

Publish (meets expectations and criteria for this Journal)

---

## Round 2 · Referee Report · Anonymous (Referee 2) · 2024-12-24

Strengths

  1. The manuscript has a good review of both BKL/mixmaster dynamics and the Carrollian limit in sections 2 and 3.
  2. The presentation is clear, and the main conceptual point on the relationship between Carrollian and BKL physics is very elegant.

Weaknesses

  1. Most of the paper seems to be review. The novel technical material is localized to a short section 4.

Report

I think this paper contains very interesting observations and deserves to be published. The impact would be greatly increased if the technology built up around the Carrollian limit allows for a technically sound and tractable framework for computing quantum and stringy corrections to BKL physics.

Requested changes

  1. I think it would be useful, in the discussion, to make some comments on the exact relationship between the ultralocality of the BKL setup and the ultralocality of the Carrollian limit. Naively the physics behind the two regimes is very similar - when approaching a spacelike singularity, light simply does not have much time to propagate, so it makes sense that the c ->0 limit is relevant. It was unclear to me whether the authors are claiming that the physics of the Kasner/BKL regime really is equivalent to the Carrollian regime, and whether it's obvious that perturbative corrections in c would accurately capture realistic corrections to BKL behavior away from the ultralocal limit.

Recommendation

Publish (meets expectations and criteria for this Journal)

---

## Editorial Decision

published